# Research

health and disease and epidemiology

physical impairment, fitness, white-footed mouse, *Peromyscus leucopus*, survival, ectoparasites

**Authors for correspondence:**
Francesca I. Rubino
e-mail: firubino@ucdavis.edu
Kelly Oggenfuss
e-mail: oggenfussk@caryinstitute.org
Richard S. Ostfeld
e-mail: ostfeldr@caryinstitute.org

# Effects of physical impairments on fitness correlates of the white-footed mouse, *Peromyscus leucopus*

Francesca I. Rubino[1], Kelly Oggenfuss[2] and Richard S. Ostfeld[2]

[1]Department of Medicine and Epidemiology, School of Veterinary Medicine, University of California, Davis, CA, USA
[2]Cary Institute of Ecosystem Studies, Millbrook, NY, USA

(iD) FIR, 0000-0001-5571-5298

Physical impairments are widely assumed to reduce the viability of individual animals, but their impacts on individuals within natural populations of vertebrates are rarely quantified. By monitoring wild populations of white-footed mice over 26 years, we assessed whether missing or deformed limbs, tail or eyes influenced the survival, body mass, movement and ectoparasite burden of their bearers. Of the 27 244 individuals monitored, 543 (2%) had visible physical impairments. Persistence times (survival) were similar between mice with and without impairments. Mice with eye and tail impairments had 5% and 6% greater mass, respectively, than unimpaired mice. Mice with tail impairments had larger home ranges than did unimpaired mice. Burdens of black-legged ticks (*Ixodes scapularis*) were higher among mice with tail and limb impairments while burdens of bot fly larvae (*Cuterebra*) were higher among mice with cataracts compared to mice without impairments. Our findings do not support the presupposition that physical impairments reduce viability in their bearers and are inconsistent with the devaluation of impaired individuals that pervaded early thinking in evolutionary biology.

## 1. Introduction

Physical impairments can be caused by trauma, disease or inheritance and may affect the behaviour, health and longevity of the individual. The perception that impairment inhibits individual fitness has influenced the management of wild animals [1–3] and multiple eugenic movements [4–9]. Despite a long history of assumptions regarding both the nature and consequences of impairments, empirical studies comparing impaired and unimpaired individuals within animal populations are rare. Analysing the impact of impairment in wild animal populations can serve as a model to improve our basic understanding of the influence of impairment on survival and fitness more generally.

Although several studies have explored the impact of impairment on non-human animals in laboratory and field settings, the applicability of such studies to inform how impairments influence fitness in the wild is limited. Studies conducted in laboratories often induce impairment and fail to replicate the challenges animals may encounter in the wild such as predation or density-dependent competition [10–14]. Many of these studies have focused on limb loss in arthropods, showing contradictory effects of impairment on reproduction [15,16], survival [12,17], competition [18] and movement [19]. Among those documenting impaired mammals in the wild, studies have often not focused on the effects of impairment on survival or other fitness measures [13] or have been limited by relatively small sample sizes [20–23]. Field studies on lizards, spiders, insects and mice have provided much larger sample sizes but have focused on species that exhibit autotomy [17,18,24–30]. This evolved trait enabling animals to self-amputate an appendage or limb represents a special case in which specific adaptations (vasoconstriction, segmentation) might minimize the acute consequences of appendage loss or damage. Autotomy has been documented in at

least 35 species of mice but not *Peromyscus leucopus*, the white-footed mouse [27]. With large wild populations and a well-studied life history, this species provides a rare opportunity to gain insight into impairment in species with no apparent adaptations to facilitate autotomy.

*Peromyscus leucopus* are resilient to parasite loads and to habitat loss and degradation. Population density is often higher in degraded forests and fragmented landscapes than in less disturbed, more continuous forest (reviewed by Ostfeld [31]). Survival of individuals heavily parasitized by *Cuterebra* sp. (bot fly) larvae and *Ixodes scapularis* (black-legged ticks) is at least as long as that of individuals with light or zero parasite burdens [32,33], and individuals with high titres of the Lyme disease bacterium, *Borrelia burgdorferi*, cannot be differentiated behaviourally or ecologically from those with low titres [34]. These observations suggest a syndrome of high physiological tolerance to damage from parasites, pathogens and environmental conditions that would appear otherwise unfavourable. Determining whether this tolerance might extend to physical impairments should contribute to our understanding of both the causes and consequences of ecological resilience in this widespread, ecologically well-connected species.

The objective of our study was to measure the effect of impairments on markers of fitness. To study this question in wild populations, we focused on the impact of visible, physical impairments detectable during the course of a long-term mark–recapture study. The effects of tail, limb and eye impairments on *Peromyscus leucopus* survival, body mass, movement and burden of ectoparasites (larval bot fly and ticks) were analysed using 26 years of data from a live-trapping programme.

## 2. Material and methods

### (a) Field approach and impairment types

Data were obtained from a small mammal mark–recapture trapping programme on property at the Cary Institute in Millbrook, New York. Six 2.25-ha plots (150 m by 150 m), two initiated in 1991 with an additional four added in 1995, were established in eastern deciduous forest dominated by oaks, *Quercus*, and maples, *Acer* (here on referred to as GX, GC, HX, HC, TX and TC). The trapping programme has been described at length [32,35]. In brief, trapping was conducted every 3–4 weeks over 2–3 consecutive days between May and November using Sherman live traps baited with oats. Small mammals were given metal identification ear-tags upon first capture, and data on sex, age (based upon pelage), mass, ectoparasite load and location (trapping station) on the plots were recorded for each capture.

Impairments were recorded in mice based upon trapping notes from 1991 to 2016. Although this was not the main aim of the data collection, established protocols instructed all trappers to record detailed notes about physical features of each mouse in the notes, including impairments. Tail impairments included missing, partially missing or broken tails. During some trapping years, tail snips (1 mm of the distal end) were collected from mice. Mice missing only what was removed during a snip were not included as tail impaired. Limb impairments included missing, partially missing, or broken/deformed limb(s). Eye impairments included missing eye(s) or cataracts. Other physical deformities which are more ambiguous in effect such as missing toes, damaged external ears or physical injuries described vaguely in the trapping notes were not included. Eleven mice had both a tail impairment and either a limb or eye impairment. Due to a small sample size, the effect

of multiple impairments could not be analysed. Instead, these mice were treated as having either limb or eye impairments as such impairments were rarer than tail impairments, which comparatively were common. The exposure of interest (impairment) was distinguished from injuries (which sometimes result in impairment) as a visible physical disfiguration acquired prior to the trapping event rather than acute damage that either healed and left no visible physical disfiguration or resulted in death.

We sought to assess the consequences of impairments for metrics of individual fitness. We did not experimentally impose impairments on mice but instead relied on correlations between impairments and fitness metrics; such a correlative approach can limit the strength of inference about causation. For each analysis, we took care to examine whether the impairments might plausibly be a consequence, rather than a cause, of differences in fitness metrics. These specific efforts to assess the potential for reverse causality are described under each analysis. Associations between sex and impairment as well as fluctuations in the number of impaired mice each year were assessed by a chi-squared test and a Pearson correlation test, respectively. Capture probability was assumed to be uniform between impaired and non-impaired mice.

All impairments were analysed as a categorical effect by general body part (no impairment, tail, limb and eye) as well as by the specific category (no impairment, missing tail, partially missing tail, broken tail, missing limb(s), partially missing limb(s), broken limb(s), missing eye(s) and cataracts). When possible, models were analysed with an additional level within the categorical effect for impairment that captured 'future impaired' to describe data from mice captured in an unimpaired state who later became impaired ($N = 301$). Variables with plausible mechanisms to influence survival, body mass, movement and ectoparasite burden were selected *a priori*. A change-in-estimate analysis was used to evaluate confounding. Model selection was based upon the smallest AIC. All analyses were conducted in R v. 3.6.3.

### (b) Survival analysis

The survival of individual mice was estimated by their persistence time on the trapping plot (as in Burns *et al.* [32]). A Cox proportional hazards model was used to compare the persistence times of impaired versus non-impaired mice employing the survival package [36]. To capture the effect of being impaired on persistence and control for the confounding effects of age prior to impairment, impaired mice were individually matched to all available controls meeting the selection criteria (age, week of first capture and minimum persistence as described below). This approach was derived from epidemiological methodologies commonly employed in human injury research [37–39]. Initiation of the impairment period for an individual was estimated as beginning at the last documented trapping event when the mouse was not yet recorded as impaired or when the impairment occurred if the timing of the causative event was known. The time between a mouse's first capture event and the start of this impairment period was considered the pre-impairment period. Mice who had no documented impairments were matched to impaired mice by age at and week of first capture. Only mice who persisted at least one trapping event longer than the respective pre-impairment period for their matched case were included as controls. Thus, the matched mice were subject to the same length of time on the plot prior to the exposure of interest, thereby minimizing the reverse causal effect in the model of age prior to impairment leading to an increased likelihood of impairment. To control for non-residents, only mice who persisted longer than one week on the trapping plot were included in the analysis. Mice who persisted until the last trapping week for each plot of 2016 were censored. No residual confounding after matching was detected for season, year and age during which the mouse was first trapped. Sex was analysed as a confounder in the model.

## (c) Body mass analysis

A linear mixed-effects model using the 'lmer' function in the lme4 package [40,41] was used to assess differences in mass associated with impairment. Sex, season and their interaction were assessed as fixed-effects and plot, individual and year as random-effects. Only individuals with adult pelage who were above 11 g and less than or equal to 40 g were included to control for potential errors in the data.

We were aware that any differences in body mass between impaired and non-impaired mice could in fact reflect a reverse causality whereby mice who are heavier were more likely to become impaired. A matched case–control design was applied, and conditional logistic regression was used to assess the directionality of the observed associations between impairment and mass. Mice who had data prior to impairment ('pre-impaired' mice) were matched to as many non-impaired mice as possible on sex, age, week and plot. Matching on week was considered necessary due to the confounding effects of time from stressors such as intra-specific competition and environmental changes. Like the survival analysis, the final trapping event for each impaired mouse before they were labelled impaired in the dataset was used to represent the pre-impaired measurements. Four of the impaired mice had no unimpaired mice available in the same trapping week and thus matched controls were used from the prior or following week from another plot (only one plot was trapped per week).

## (d) Movement analysis

The movement of individual mice was quantified as the mean-squared distance (MSD) around the centre of activity during a season within a plot [42]. The MSD was calculated by first assigning a coordinate grid to each plot. The centre of activity was estimated by triangulating distances between all the traps the mouse was captured at and finding the mouse's mean coordinates on the grid. The distance between each trap and the estimated centre of activity was calculated, those distances were squared to compute the area travelled, and finally, the squares were averaged to compute the MSD travelled as a measure of movement. The analysis was restricted to mice considered residents with sufficient data, defined as mice who were trapped at least three times within a season on the same plot in the same year [43]. Some mice changed impairment status during the seasonal period; the MSDs of these mice were coded as the final impairment status during the period as there was limited knowledge on when mice received their impairments. Mice whose centre of activity was near the edge of a grid may appear to travel less simply due to travelling more frequently outside of the grid. However, being on the edge of the grid (defined as within 15 m) was not associated with impairment and thus was not considered a confounder. A linear mixed model was used to analyse the association between impairment and the calculated movement level, controlling for the fixed-effects of sex, season, their interaction, and age, as well as the random-effects of plot, individual and year. Although the MSD did not need to be normally distributed [44], the strong skew towards high MSD suggested that model fit would be improved if MSD was natural log transformed (as was done in Klein & Cameron [45]).

## (e) Ectoparasite analyses

The ectoparasite burden for larval ticks (mainly *Ixodes scapularis*) and bot flies were quantified by ectoparasite load and by infestation risk. The relationship between impairment and whether a mouse was infested was assessed by a logistic mixed-effects model for each parasite type controlling for the fixed-effects of season, sex, their interaction, age and mass and the random-effects of individual, plot and year. Negative binomial models were created for the same variables to analyse load. Among

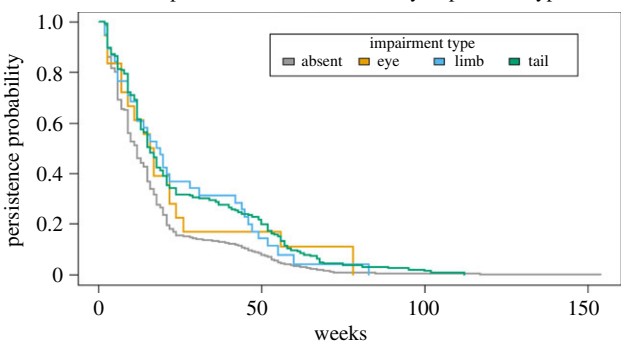

**Figure 1.** The Kaplan–Meier survival curves for 3096 mice show that most mice with impairments had an elevated persistence probability compared to non-impaired mice, and tail-impaired mice persisted longer than limb or eye impaired, before controlling for confounders. (Online version in colour.)

mice with at least one bot fly larva, the relationship between impairment and level of infestation was analysed using a zero-truncated negative binomial model. Zero-inflation was ruled out *a priori*. The reduced performance of the model using its zero-inflated version was confirmed using the Vuong test.

It has previously been shown that counts of larval ticks on the head and ears correspond well to the true tick load [46], whereas the nymphal tick burden can be harder to quantify in the field. Thus, only larval ticks were included in the analysis. To capture the tick season, data collected from April to October were included in the tick models. Data from 1991 to 1992 were excluded due to a lack of information. Season was grouped into a two-level variable as low (mean tick burden less than 10) and high (mean tick burden greater than 10, August and September).

Similarly, the bot fly larvae analysis was limited to the larval bot fly season, defined as the range of weeks each year from when the first larval bot fly infestation was recorded until the last week infestation was recorded. Seasonal effects were also dichotomized as high (greater than 10% of mice infested, August and September) versus low (less than 10% of mice infested).

## 3. Results

Over 26 years, 543 (1.99%) of 27 244 individual mice were recorded with visible physical impairments of the eyes or major appendages (electronic supplementary material, table S1). The number of impaired mice correlated significantly ($r(24) = 0.86$, $p < 0.001$) with the overall number of mice over time (electronic supplementary material, figure S1). Sex was not associated with impairment ($X^2$ (1, $N = 27\,130$) = 0.55, $p = 0.47$) nor impairment type (tail: $X^2$ (1, $N = 27\,015$) = 0.72, $p = 0.40$; limb: $X^2$ (1, $N = 26\,658$) = 1.67, $p = 0.20$; eye: $X^2$ (1, $N = 26\,635$) = 2.43, $p = 0.12$). The documented percentage of all impairment types increased with age (electronic supplementary material, table S2).

### (a) Survival

In total, 233 mice with impairments were matched with 2863 mice without impairments. The average persistence time in the entire cohort used in the analysis was 17.23 weeks, ranging from 2 to 154 weeks. No evidence for an association between impairment and a reduction in survival was found. No statistically significant differences in median survival times (electronic supplementary material, table S3; figure 1) were detected between impaired and non-impaired mice ($X^2 = 2.3$, $p = 0.1$) nor across impairment types and non-impaired mice

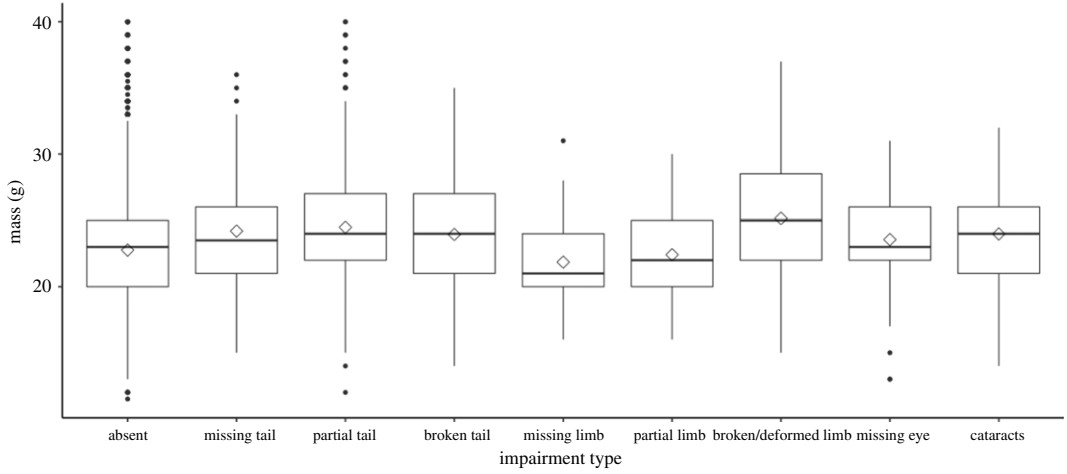

**Figure 2.** Boxplot of adult mass by impairment type for mice trapped between 1991 and 2016. Mean represented by diamond symbol.

$(X^2 = 3.0, p = 0.4)$. After controlling for sex, impairment (tail hazard ratio: 0.88, 95% CI: 0.73–1.05%; limb: 0.82, 95% CI: 0.55–1.23; eye: 0.99, 95% CI: 0.57–1.73) was not strongly associated with longer persistence times in the Cox proportional hazards model (electronic supplementary material, table S4).

## (b) Mass

On average, adult mice without impairments had less mass (22.76 g ± 4.05) than mice with impairments (electronic supplementary material, table S5; figure 2). The highest mean mass was observed among mice with tail impairments (mean = 24.33 g ± 4.27), followed by eye impairments (23.84 g ± 4.06), and limb impairments (22.83 g ± 4.27). Season, sex and their interaction improved fit for the mixed-effects models (table 1; electronic supplementary material, table S6). After adjustment, tail and eye impairments were significantly associated with having on average 1.44 g (95% CI: 1.09–1.79, $p < 0.001$) and 1.09 g (95% CI: 0.01–2.18, $p = 0.05$) more mass, respectively, than mice without impairments. In proportion to the average adult's mass in the study, tail and eye impairments were associated with 6% and 5% higher masses, respectively. More specifically, after controlling for the selected variables having a missing tail was associated with having on average 2.14 g more mass (95% CI: 0.84–3.43, $p = 0.001$), a partially missing tail with having on average 1.45 g more mass (95% CI: 1.03–1.87, $p < 0.001$), a broken tail with having on average 1.21 g more mass (95% CI: 0.51–1.91, $p = 0.001$) and cataracts with having on average 1.64 g more mass (95% CI: 0.24–3.03, $p = 0.02$). No association between mass and limb impairment was detected (−0.03 g, 95% CI: −0.96–0.90, $p = 0.95$).

In the matched analysis of the impact of body mass on the odds of becoming impaired, mice were matched as follows: 224 mice prior to tail impairment to 6382 controls (mice without a recorded impairment), 40 mice prior to limb impairment to 999 controls and 23 mice prior to eye impairment to 550 controls. Due to small sample sizes, the analysis was not conducted on specific impairment categories. Matching did not appear to introduce bias as no improvement to the conditional logistic model was detected for any of the selection factors. Overall, no relationship was observed between a mouse's mass and the odds of becoming impaired (OR = 1.00, 95% CI: 0.97–1.04, $p = 0.84$). Similarly, no relationship was observed for each category of impairment

**Table 1.** Results of four-level mixed-effects linear regression examining the association between general impairment types and body mass. Number of observations = 52 724 and number of individual mice = 18 765.

| parameter (reference) | estimate | 95% CI | | p-value |
|---|---|---|---|---|
| intercept | 22.78 | 22.43, | 23.13 | <0.001 |
| impairment (absent) | | | | |
| tail | 1.44 | 1.09, | 1.79 | <0.001 |
| limb | −0.03 | −0.96, | 0.90 | 0.95 |
| eye | 1.09 | 0.01, | 2.18 | 0.05 |
| sex (female) | | | | |
| male | −0.81 | −0.93, | −0.69 | <0.001 |
| season (fall) | | | | |
| spring | 0.50 | 0.37, | 0.63 | <0.001 |
| summer | −1.91 | −1.99, | −1.83 | <0.001 |
| interaction sex and season | | | | |
| male: spring | −0.31 | −0.50, | −0.12 | 0.001 |
| male: summer | 1.03 | 0.92, | 1.15 | <0.001 |
| **random effects** | **variance** | **s.d.** | | |
| tag | 8.84 | 2.97 | | |
| grid | 0.02 | 0.13 | | |
| year | 0.71 | 0.84 | | |

(tail: OR = 1.02, 95% CI: 0.98–1.06, $p = 0.33$; limb: OR = 0.96, 0.87–1.05, $p = 0.37$; eye: OR = 0.95, 0.82–1.10, 0.45) (electronic supplementary material, figure S2).

## (c) Movement

On average, home ranges of individual mice extended 256.1 m$^2$ from centres of activity within a season. Mean MSD among mice with tail impairments was highest overall, measuring 20.16% greater than the average among controls who had no recorded impairments throughout the trapping data (electronic supplementary material, table S7 and figure S3). Mice who became impaired at a later time but who were not yet impaired moved on average 10.79% farther than controls. Age, sex and season all improved the final

**Table 2.** Results of the mixed-effects linear regression examining the association between general impairment types and the log(MSD). Number of observations = 14 231 and number of individual mice = 10 252.

| parameter (reference) | coefficient | 95% CI | | p-value |
|---|---|---|---|---|
| intercept | 5.24 | 5.00, | 5.48 | <0.001 |
| impairment (absent) | | | | |
| tail | 0.29 | 0.10, | 0.48 | 0.002 |
| limb | 0.05 | −0.48, | 0.59 | 0.85 |
| eye | −0.03 | −0.67, | 0.60 | 0.92 |
| future impaired | 0.20 | 0.03, | 0.37 | 0.02 |
| sex (male) | | | | |
| female | −0.41 | −0.46, | −0.36 | <0.001 |
| season (fall) | | | | |
| spring | 0.22 | 0.12, | 0.33 | <0.001 |
| summer | 0.18 | 0.13, | 0.23 | <0.001 |
| age (adult) | | | | |
| juvenile | −0.13 | −0.22, | −0.03 | 0.008 |
| subadult | −0.15 | −0.21, | −0.08 | <0.001 |
| **random effects** | **variance** | **s.d.** | | |
| individual | 0.39 | 0.63 | | |
| plot | 0.01 | 0.08 | | |
| year | 0.32 | 0.57 | | |

**Table 3.** Results of the negative binomial mixed-effects model examining the association between general impairment types and bot fly larvae load. Number of observations: 50 917; number of individuals: 20 468. IRR = incidence rate ratio.

| parameter (reference) | IRR | 95% CI | | p-value |
|---|---|---|---|---|
| intercept | 0.03 | 0.02, 0.05 | | <0.001 |
| impairment (absent) | | | | |
| tail | 1.10 | 0.93, 1.31 | | 0.27 |
| limb | 0.85 | 0.48, 1.50 | | 0.57 |
| eye | 2.09 | 1.30, 3.37 | | 0.002 |
| future tail impaired | 0.78 | 0.59, 1.02 | | 0.07 |
| future limb impaired | 1.34 | 0.72, 2.48 | | 0.36 |
| future eye impaired | 1.41 | 0.70, 2.83 | | 0.34 |
| season (low) | | | | |
| high | 5.47 | 5.14, 5.83 | | <0.001 |
| age (adult) | | | | |
| juvenile | 0.48 | 0.41, 0.56 | | <0.001 |
| subadult | 0.60 | 0.55, 0.66 | | <0.001 |
| sex (female) | | | | |
| male | 1.19 | 1.13, 1.26 | | <0.001 |
| **random effect** | **variance** | **s.d.** | | |
| individual | 0.30 | 0.55 | | |
| plot | 0.05 | 0.22 | | |
| year | 0.64 | 0.80 | | |

models on movement. After adjustment (table 2; electronic supplementary material, table S8), tail impairments were associated with an estimated increase in mean MSD of 33.82% compared to non-impaired mice ($p = 0.002$). More specifically, mice with partially missing tails had on average a 42.21% higher MSD than controls ($p = 0.003$). Mice who were not yet impaired but became impaired later in their lives had 22.05% higher MSDs on average than controls ($p = 0.02$). Although mice with eye impairments and mice with limb impairments had higher MSDs on average within a season compared to controls, after controlling for confounding, these differences were not statistically significant.

## (d) Ectoparasite burden
The median larval tick burden among mice was 2 for those without visible physical impairments (IQR:0–7) and those with either tail (IQR:0–9) or limb impairments (IQR:0–9) and 4 for those with eye impairments (IQR:0–10). After controlling for season, sex, their interaction, and age, no significant associations were detected between impairments and the odds of tick infestation (tail: OR = 0.99, 95% CI: 0.84–1.16, $p = 0.87$; limb: OR = 0.91, 95% CI: 0.58–1.42, $p = 0.68$; eye: OR = 0.87, 95% CI: 0.51–1.50, 0.62). However, higher tick loads were associated with tail impairments after controlling for the same variables (tail: IRR = 1.11, 95% CI: 1.01–1.22, $p = 0.04$). No association between tick load and eye or limb impairments overall were detected (limb: OR = 1.14, 95% CI: 0.89–1.47, $p = 0.30$; eye: OR = 1.19, 95% CI: 0.88–1.61, 0.26). A modest association between a higher tick load and missing a limb was observed in the model analysing specific impairments (IRR = 1.51, 95% CI: 0.99–2.29, $p = 0.05$). Zero-

inflation was not detected. Full descriptive statistics and model results are reported in the electronic supplementary material, tables S9–13.

During the bot fly season, the number of bot fly larvae per mouse ranged from 0 to 9. The per cent of mice infested with at least one bot fly ranged from 11.11% of mice with limb impairments, 12.44% of mice who became impaired at a later date, 13.05% of mice without visible physical impairments, 17.11% of mice with tail impairments and 18.63% of mice with eye impairments (electronic supplementary material, table S14). After controlling for season, sex and age in the logistic mixed-effects models, a significant association was observed between the odds of having at least one bot fly larvae and having an eye impairment (OR = 1.81; 95% CI: 1.01–3.25; $p = 0.05$). More specifically, mice with cataracts had 2.19 times the odds (95% CI: 1.04–4.62) as mice without visible physical impairments of being infested with bot fly larva ($p = 0.04$). No significant association was detected for a tail or limb impairment, nor among mice who became impaired at a later date, summarized in electronic supplementary material, tables S15–16 (tail: OR = 1.08, 95% CI: 0.88–1.31, $p = 0.46$; limb: OR = 0.89, 95% CI: 0.48–1.63, $p = 0.70$; future impaired: OR = 0.86, 95% CI: 0.65–1.12, 0.26). The negative binomial models showed similar results for bot fly load after controlling for the same factors (table 3; electronic supplementary material, tables S17–18). Mice with cataracts had 2.57 times more bot fly larvae than mice without impairments (95% CI: 1.43–4.63, $p = 0.002$). To identify whether the higher load among eye impaired mice

preceded impairment, mice who became impaired at a later date were separated by their impairment type and analysed separately. Compared to those who did not have a recorded impairment, no significant differences were detected between mice who were recorded to have developed impairments at a later time point and controls (future tail: OR = 0.78, 95% CI: 0.59–1.02, $p = 0.07$; future limb: OR = 1.34, 95% CI: 0.72–2.48, $p = 0.36$; future eye: OR = 1.41, 95% CI: 0.70–2.83, 0.34). Due to small sample sizes, the association was only analysed at the general impairment scale.

## 4. Discussion

We analysed the associations between tail, limb and eye impairments and *Peromyscus leucopus* survival, body mass, home range size and ectoparasite burdens using 26 years of mark–recapture data on 27 244 individuals, of whom 544 had impairments. Simple expectations were that impairments might make mice more vulnerable to predators, thus reducing survival, reduce locomotor efficiency and thus food intake, body mass and home range size, and reduce detection and/or grooming efficiency, thus increasing ectoparasite burdens. Although we found limited associations between physical impairments and increased ectoparasite burden, we found no evidence that physical impairments in *Peromyscus leucopus* overall were associated with a decrease in metrics of fitness. On the contrary, mice with impairments had similar persistence times, higher body masses for those with tail or eye impairments, and larger home ranges for those with tail impairments, on average, than mice without impairments.

The larger home ranges of mice with tail impairments and similar home ranges of mice with limb or eye impairments compared to mice without impairments suggest that horizontal movement was not impeded by impairment. Larger home ranges may be linked to behavioural differences in impaired versus non-impaired mice. For instance, if impaired mice are less risk averse, they may use more space in daily activities; this behaviour may have also increased their risk of becoming injured from predators or agonistic intra-specific interactions which led to their impairment.

We observed a relationship between tick load and missing a limb or having a tail impairment. The large home ranges of mice with tail impairments may have increased the probability of encountering host-seeking ticks (which are sedentary and rely on hosts to approach them). Potentially, limb-impaired mice could have reduced grooming efficiency. The trend toward higher tick burdens on tail- and limb-impaired mice may reflect one or both mechanisms, although the effect seems modest. Previous work on louse burden and impairment may illuminate why mice with limb impairments did not have even higher tick burdens. Lodmell et al. [11] found that in the laboratory setting louse populations on mice with induced limb amputations remained low if the mice remained housed together and thus able to groom one another. The role intra-specific social interactions play in the success of impaired individuals was beyond the scope of this study, but likely influences the implications of these findings for other organisms.

Although mice with eye impairments had a higher bot fly larvae burden on average than did unimpaired mice, previous research by Burns et al. [32] and Cramer and Cameron [47] showed that bot fly parasitism is associated with increased survival and body condition of *Peromyscus leucopus*. Burns et al. [32] did, however, observe negative associations between bot fly parasitism and reproductive activity, a correlate of fitness we did not assess. Increased bot fly larvae burden and cataracts were most strongly linked. Mice with eye impairments may spend more time on the ground where they may come into more frequent contact with bot fly eggs. Susceptibility to larval bot flies and to cataracts may share a common cause, such as susceptibility to infections, or lax grooming behaviour. Additionally, the initial stress from an injury that causes impairment may be linked to increased parasitism. Stressful stimuli are associated with an increase in parasite burden in many species [48–53] including *Peromyscus leucopus* [54]. However, no relationship was observed between mice prior to their eye impairments and larval bot fly burden, providing evidence against such a predisposition in the populations we studied.

Our findings of similar persistence and higher average mass among tail-impaired mice in this species were in concordance to the findings in Shargal et al. [27] on *Acomys* spp. (spiny mice) which exhibit caudal autotomy. Although *Peromyscus leucopus* do not themselves exhibit caudal autotomy, it is plausible that similar strategies in anti-predator behaviour may lead to tail loss and increased probability of surviving a predator attack. As no statistically significant change in mass was detected in the matched analysis, the relationships described between impairment and mass do not appear likely to be due to mice with more mass having a higher risk of becoming impaired but rather due to mice who are impaired being able to maintain more mass.

We detected physical impairments in field populations of mice that were repeatedly captured via a live-trapping protocol. The impairments that we could detect in a few moments of inspection in the field were discrete and obvious to observers. More subtle or internal impairments were likely missed. We recognize the likely existence of classes of mice—those that were killed by predators, competitors, or pathogens as a result of their impairment, and those that died as an immediate result of injury—that we failed to detect. Nevertheless, by comparing fitness correlates of mice with visible impairments to those of mice with no visible impairments, we were able to directly assess impacts of impairments on fitness in the weeks and months following the impairment. Moreover, by comparing cohorts of mice that later became impaired with those that did not, our analyses constitute a fair assessment of fitness consequences of specific, visible impairments.

There were several limitations in this study. First, as we did not know the exact time each mouse spent impaired, we could not adjust for possible adaptation periods or effects of age at which the animal became impaired. These factors may have influenced individual results. Carey et al. [13] found that the location, quantity and age of impairment among artificially impaired *Drosophila* modified the impacts of impairment on life expectancy. Additionally, survivorship bias was of particular concern as is common with other retrospective studies. We only included mice who survived injuries long enough to become impaired rather than also including acutely injured individuals. However, this bias was limited in our analysis as the exposure under study was impairment rather than the effect of injuries and thus by definition the individual needed to survive until an injury developed into a visible physical impairment. Finally, we did not analyse the association between impairment

and fecundity. Further research is needed to evaluate these associations and identify if the relationships we have documented are associated with trade-offs in fecundity.

Overall, the results of our long-term studies on wild *Peromyscus leucopus* do not support the presupposition, sometimes applied to vertebrates more broadly [1,2,4], that physical impairments such as missing or damaged limbs and eyes reduce measures of fitness such as survival, movement and mass in their bearers. Our finding that impaired mice exhibited equal or greater viability in the fitness factors we measured is inconsistent with the devaluation of impaired individuals that originated in early thinking in evolutionary biology and that has persisted in some more modern applications [2,4,7,9].

The results of this study reveal a high level of resilience of *Peromyscus leucopus* to a suite of physical impairments that might be expected to compromise movement, predator avoidance, resource acquisition and self-protection. Mice appear able to compensate for broken or missing major appendages and for partial or total loss of vision in ways that generally avoid compromising longevity, body condition, movement or protection from ectoparasites. Such resilience is consistent with prior studies revealing a syndrome of tolerance to infection by some microparasites [34] and ectoparasites [32,33]. Whether such a syndrome of tolerance to injury and infection is shared with other species, and if so, what mechanisms might underlie high tolerance, are unknown, but worthy of further study.

Ethics. This study was conducted under Cary Institute Institutional Animal Care and Use Committee protocols and conformed to the ASM guidelines for the use of wild mammals in research.

Data accessibility. Data are available from the Dryad Digital Repository: https://doi.org/10.25338/B8P62H [55]. The data are provided in the electronic supplementary material [56].

Authors' contributions. F.I.R.: conceptualization, formal analysis, investigation, methodology and writing-original draft; K.O.: data curation; R.S.O.: data curation, formal analysis, investigation, writing-original draft, writing-review and editing. All authors gave final approval for publication and agreed to be held accountable for the work performed therein.

Competing interests. The authors have no competing interests to declare.

Funding. This work was supported by grant nos. DEB 0075277, DEB 0444585, DEB 0949702 and DEB 1456527 from the Long Term Research in Environmental Biology (LTREB) programme of the United States National Science Foundation (NSF).

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
