## [Peer Review File · Proceedings of the Royal Society B: Biological Sciences]

Review History

RSPB-2021-0886.R0 (Original submission)

Review form: Reviewer 1

Recommendation

Accept with minor revision (please list in comments)

Scientific importance: Is the manuscript an original and important contribution to its field?

Good

General interest: Is the paper of sufficient general interest?

Good

Quality of the paper: Is the overall quality of the paper suitable?

Good

Is the length of the paper justified?

Yes

Should the paper be seen by a specialist statistical reviewer?

No

Do you have any concerns about statistical analyses in this paper? If so, please specify them explicitly in your report.

Yes

It is a condition of publication that authors make their supporting data, code and materials available - either as supplementary material or hosted in an external repository. Please rate, if applicable, the supporting data on the following criteria.

Is it accessible?

Yes

Is it clear?

Yes

Is it adequate?

Yes

Do you have any ethical concerns with this paper?

No

Comments to the Author

This paper is well written, and is generally a contribution to the literature, but needs some work.

Line 24: need to include Latin binomial or at least genus for "tick and bot fly"

Line 36: "Natural Selection" need not be capitalized

Line 41: comma after "thus"

Lines 36-46: The title and abstract mention and specifically speak about mice. I think when talking about "individuals" in these opening lines, you need to make clear you are talking about humans. Lines 41 and 44, change "individuals" to "humans."

Lines 47-48: Do you really think, and expect the readership to believe, that the role impairments play in humans can be better understood by looking at those in a single mouse population found on Cary Institute property? I would say this is a bit of stretch. And you are assuming that mice are born with these impairments and not sustaining them from predator encounters or from human sampling events to be able to compare to human birth defects? This opening paragraph is impossibly broad, comparing human eugenics to a mouse who may have lost its tail which saved its life in an attack from a barred owl. Your Abstract is spot on, using these data to attribute survival of the fittest within species is an addition to the literature base; pulling from and comparing to human eugenics is an enormous over-reach.

Line 92: oak and maple should at least have the genus identified

Lines 94-95: I know this is a brief description and you reference more specific methods, but your description here seems to indicate you trapped mice every 3-4 weeks since 1991. I am guessing you were not trapping in January, so might want to include that you were trapping every 3-4 weeks from May - October, or whatever the range of months that might be.

Line 159: Likely not many mice under 11 grams existed in the data set because mice below that mass were unlikely to trigger Sherman traps.

Lines 197-198, 202, 213, 318, 320, 321, 325, 327, 328, 334, 336, 340, 350, 371, 375, 378, 382: need to specify they were parasitized by bot fly larvae, not bot flies

Line 202: why is "botfly" one word here and two words everywhere else?

Line 238: "3" should be "3.0"

Line 243: "3096" should be "3,096" to remain consistent with previous usage

Line 247: comma after "average"

Line 247: "lighter" is a reference to weight. Here you are measuring mass. "On average, adult mice without impairments had less mass . . ."

Line 248: delete the comma and insert "mass" after "observed"

Lines 249-250: don't you need to swap the order of limb impaired mice and eye impaired mice if you are reporting those cohorts in descending order of average mass?

Lines 251-261: You keep saying/ implying that various impairments caused an increase in mass, but that is really not the case. Mice with these impairments had more mass, generally. Semantics, I know. And this is not surprising as elder mice (with more mass) will have had more intraspecific and interspecific conflicts than younger mice merely as a function of age.

Lines 262-263: You had multiple leucopus with 30-50 grams of mass in your study? In the thousands we have trapped over 15 years, we seldom have had one over 30 grams, and usually they were gravid females. Based on personal experience, I would have to agree with this site on the physical description of leucopus, including their mass of between 15-25 grams (https://animaldiversity.org/accounts/Peromyscus_leucopus/). I would double check these figures before publishing. They seem like they could be data errors in the "Absent, Missing Tail, and Partial Tail" box and whisker plots. Those are BIG leucopus that I have never seen reported in the literature previously. People in the know are going to question the veracity of your data if these data are published as is and are not addressed.

Lines 307-308: if you are going to include the mean ectoparasite burden for mice without impairments, you need to include it for impaired mice as well instead of just referring the to supplemental table. Do you have any statistical proof that mean burdens on impaired mice were significantly higher than without impairments? If so, here is where you need to include that in text. Otherwise, this is meaningless.

Line 310: comma after "average"

Lines 312-314 and 314-316: even if there was no significance, you need to include statistical outputs here

Lines 318-319: need to include mean burdens in text here as well instead of referencing a supplemental table

Line 319-320, 321-323: need to include statistical output showing lack of significant differences

Lines 356-365: You have this exactly backwards. The mice do not have larger home ranges because they had a tail impairment, they have tail impairments because they had larger home ranges. These mice prior to becoming impaired, were pre-programmed to wander and have larger home ranges, placing them further away from the safety of their dens than their counterparts. As a result, these mice were more exposed and more likely to encounter a predator attack, thus causing said tail impairment. Most mice with larger home ranges without tail impairments were dead as the result of predation and were not available to be trapped. This entire paragraph needs to be omitted and re-written because it is wrong and will fall completely flat if published as is.

Line 367-369: Yes!!! Omit the vertical habitat usage altogether and focus on this. Mice with larger home ranges are more exposed and thus, more at risk. This is the focus of this paragraph (not a closing example) and not that they cannot climb because of a damaged tail.

Line 372: period after "al"

Line 377: "...may come into more frequent contact with bot fly eggs."

Line 381: delete second "in"

Line 383: In lines 20 and 89, you mention you monitored a single wild population of white-footed mice. Why here then are you suggesting you monitored multiple populations?

Line 391: It is likely you saw higher parasitism on impaired mice because of their propensity to travel further than other mice, thus increasing their susceptibility to encountering both predators and parasites.

Line 395: "et al." need not be in italics here to remain consistent with the rest of the text

Line 398: comma after "study" and delete "on," or reword the sentence as I find it difficult to understand what you are trying to say.

Line 432: "developed"

Review form: Reviewer 2

Recommendation

Accept with minor revision (please list in comments)

Scientific importance: Is the manuscript an original and important contribution to its field?

Excellent

General interest: Is the paper of sufficient general interest?

Good

Quality of the paper: Is the overall quality of the paper suitable?

Excellent

Is the length of the paper justified?

Yes

Should the paper be seen by a specialist statistical reviewer?

No

Do you have any concerns about statistical analyses in this paper? If so, please specify them explicitly in your report.

Yes

It is a condition of publication that authors make their supporting data, code and materials available - either as supplementary material or hosted in an external repository. Please rate, if applicable, the supporting data on the following criteria.

Is it accessible?

Yes

Is it clear?

Yes

Is it adequate?

Yes

Do you have any ethical concerns with this paper?

No

Comments to the Author

Here, the authors use a powerful long-term dataset of wild *Peromyscus* to address a widespread convention that physical impairment impacts host fitness. They investigated limb, tail, and eye impairments on host survival, movement, weight, and ectoparasite infection and conclude that there is no support for impairments negatively impacting host movement & fitness save for moderate effects of increased bot fly infection. I found this manuscript enjoyable to read and a robust & thoughtful study. I have some comments which I feel should be clarified/ addressed before publication. I detail these below but feel they are all addressable with minor revisions and that this study is well-suited to Proceedings B.

Minor comments

Lines 23-25 in the abstract highlight results which were not supported by the models, particularly the effects on ticks and persistence. I think the phrasing in the discussion which presents the results as similar persistence times between the groups and no effect on ticks is more transparent and that these lines as they are slightly misleading for the model results.

Line 99 – It would be helpful here just to clarify/confirm whether the recording of impairments was standard recording for notes at each capture. It seems so but initially I wondered if there could be any missing impairments if this was not an aim of data collection throughout all years

trapped.

Lines 106-108 – Given the small sample size this is not much of an issue, but I wondered if authors could have three columns for tail, eye, or limb impairments as binary variables rather than having one variable for impairments and having to assign mice to one category?

Line 122-123 – Was the assumption of uniform capture probability between these two groups tested in any way? It seems that several mechanisms being tested in this study could also affect capture probability (eg movement/ fitness).

Lined 128-130 – The N of mice for which a status change of unimpaired to impaired over the course of trapping would be informative here.

Lines 138-140 – Does ‘selection criteria’ here refer only to age and week of first capture as mentioned further down in this section? If so suggest explicitly stating when first mentioning selection criteria.

Line 201 – Some justification for why the interaction of sex and season was not tested in ectoparasite models but was for other metrics tested would be useful.

Lines 201-203 – It is not clear to me from this description why negative binomial models were used for all (?) ectoparasites and then botflies were singled out for Poisson models for mice with at least one botfly.

Lines 222-223 – Clarify here what it indicates that the impaired population correlates with overall mouse population. In space? In time? Something else?

Lines 236-237 – As stated above this result comes as surprising based on the abstract. It is reasonable to present the raw data summaries throughout the results, in several instances it reads as a bit of an afterthought that there were no effects from the models accounting for other covariates, so would caution conservative phrasing of the raw data comparisons.

Lines 276-277 – This statement regarding selection bias is a bit lost on me in terms of conveying how this was determined, perhaps rephrase?

Line 316 – How was zero-inflation ruled out?

Line 330 – The use of ‘risk’ here seems in contrast following results describing effects on bot fly number as that term should pertain to probability of infection and not load, while there was no effect on odds of botfly infestation.

Lines 336-337 – Incidence rate seems inappropriate to describe data that does not include zeroes, as it suggests information about the risk/rate of cases, which requires there to be uninfected and infected individuals. Some clarification on the ‘at least one’ criteria may be needed if this does not refer to only non-zero data.

Line 405-406 – Unclear what ‘risk’ is referring to in this sentence given this is discussing body mass.

Statistical models – I had a few queries about the treatment of terms in statistical models.

1. Given the number of levels in the variable, it seems more suitable for year to be considered as a random effect.

2. It was unclear to me why grid was considered a random effect for some models but was a fixed effect for tick and movement models.
3. Was there a reason the intercept level for year changes between models for mass and movement?

Discussion – Just a general comment that I felt the authors handled the discussion very nicely and logically. I found the hypotheses regarding vertical movement and future directions very interesting and thought-provoking, particularly potential implications of social interactions affected by impairments.

Decision letter (RSPB-2021-0886.R0)

15-Jun-2021

Dear Ms Rubino:

I am writing to inform you that your manuscript RSPB-2021-0886 entitled "Effects of Physical Impairments on Fitness Correlates of the White-footed Mouse, *Peromyscus leucopus*" has, in its current form, been rejected for publication in Proceedings B.

This action has been taken on the advice of referees, who are generally positive about the work but have also recommended that revisions are necessary. With this in mind we would be happy to consider a resubmission, provided the comments of the referees are fully addressed. However please note that this is not a provisional acceptance.

Sincerely,
Professor Hans Heesterbeek
mailto: proceedingsb@royalsociety.org

Associate Editor

Comments to Author:

Following two expert reviews there is considerable enthusiasm that the manuscript submitted by Rubino et al will make a significant contribution to understanding how physical impairment affects fitness in wildlife. The reviewers raise a few issues that should be addressed in any revision. One reviewer raises the point that the comparison to the human eugenics movement, while an interesting hook for the article, is a bit of a stretch and therefore the manuscript should focus on physical impairment in wildlife as the motivating framework for the research.

Additionally, an interesting point is raised concerning cause and effect: do individuals with impairments have larger home ranges because of impairment, or would unimpaired individuals be more likely to become impaired due to having larger home ranges? Minor concerns also were raised about the heavy reliance on supplementary tables, which most readers are unlikely to read, and the need for more detailed statistical reporting in the main text. The authors should address all of the points raised by the reviewers in preparing a revision of their manuscript.

Reviewer(s)' Comments to Author:

Referee: 1

Comments to the Author(s)

This paper is well written, and is generally a contribution to the literature, but needs some work.

Line 24: need to include Latin binomial or at least genus for "tick and bot fly"

Line 36: "Natural Selection" need not be capitalized

Line 41: comma after "thus"

Lines 36-46: The title and abstract mention and specifically speak about mice. I think when talking about "individuals" in these opening lines, you need to make clear you are talking about humans. Lines 41 and 44, change "individuals" to "humans."

Lines 47-48: Do you really think, and expect the readership to believe, that the role impairments play in humans can be better understood by looking at those in a single mouse population found on Cary Institute property? I would say this is a bit of stretch. And you are assuming that mice are born with these impairments and not sustaining them from predator encounters or from human sampling events to be able to compare to human birth defects? This opening paragraph is impossibly broad, comparing human eugenics to a mouse who may have lost its tail which saved its life in an attack from a barred owl. Your Abstract is spot on, using these data to attribute survival of the fittest within species is an addition to the literature base; pulling from and comparing to human eugenics is an enormous over-reach.

Line 92: oak and maple should at least have the genus identified

Lines 94-95: I know this is a brief description and you reference more specific methods, but your description here seems to indicate you trapped mice every 3-4 weeks since 1991. I am guessing you were not trapping in January, so might want to include that you were trapping every 3-4 weeks from May - October, or whatever the range of months that might be.

Line 159: Likely not many mice under 11 grams existed in the data set because mice below that mass were unlikely to trigger Sherman traps.

Lines 197-198, 202, 213, 318, 320, 321, 325, 327, 328, 334, 336, 340, 350, 371, 375, 378, 382: need to specify they were parasitized by bot fly larvae, not bot flies

Line 202: why is "botfly" one word here and two words everywhere else?

Line 238: "3" should be "3.0"

Line 243: "3096" should be "3,096" to remain consistent with previous usage

Line 247: comma after "average"

Line 247: "lighter" is a reference to weight. Here you are measuring mass. "On average, adult mice without impairments had less mass . . ."

Line 248: delete the comma and insert "mass" after "observed"

Lines 249-250: don't you need to swap the order of limb impaired mice and eye impaired mice if you are reporting those cohorts in descending order of average mass?

Lines 251-261: You keep saying/ implying that various impairments caused an increase in mass, but that is really not the case. Mice with these impairments had more mass, generally. Semantics, I know. And this is not surprising as elder mice (with more mass) will have had more intraspecific and interspecific conflicts than younger mice merely as a function of age.

Lines 262-263: You had multiple leucopus with 30-50 grams of mass in your study? In the thousands we have trapped over 15 years, we seldom have had one over 30 grams, and usually they were gravid females. Based on personal experience, I would have to agree with this site on the physical description of leucopus, including their mass of between 15-25 grams (https://animaldiversity.org/accounts/Peromyscus_leucopus/). I would double check these figures before publishing. They seem like they could be data errors in the "Absent, Missing Tail, and Partial Tail" box and whisker plots. Those are BIG leucopus that I have never seen reported in the literature previously. People in the know are going to question the veracity of your data if these data are published as is and are not addressed.

Lines 307-308: if you are going to include the mean ectoparasite burden for mice without impairments, you need to include it for impaired mice as well instead of just referring the to supplemental table. Do you have any statistical proof that mean burdens on impaired mice were significantly higher than without impairments? If so, here is where you need to include that in text. Otherwise, this is meaningless.

Line 310: comma after "average"

Lines 312-314 and 314-316: even if there was no significance, you need to include statistical outputs here

Lines 318-319: need to include mean burdens in text here as well instead of referencing a supplemental table

Line 319-320, 321-323: need to include statistical output showing lack of significant differences

Lines 356-365: You have this exactly backwards. The mice do not have larger home ranges because they had a tail impairment, they have tail impairments because they had larger home ranges. These mice prior to becoming impaired, were pre-programmed to wander and have larger home ranges, placing them further away from the safety of their dens than their counterparts. As a result, these mice were more exposed and more likely to encounter a predator attack, thus causing said tail impairment. Most mice with larger home ranges without tail impairments were dead as the result of predation and were not available to be trapped. This entire paragraph needs to be omitted and re-written because it is wrong and will fall completely flat if published as is.

Line 367-369: Yes!!! Omit the vertical habitat usage altogether and focus on this. Mice with larger home ranges are more exposed and thus, more at risk. This is the focus of this paragraph (not a closing example) and not that they cannot climb because of a damaged tail.

Line 372: period after "al"

Line 377: "...may come into more frequent contact with bot fly eggs."

Line 381: delete second "in"

Line 383: In lines 20 and 89, you mention you monitored a single wild population of white-footed mice. Why here then are you suggesting you monitored multiple populations?

Line 391: It is likely you saw higher parasitism on impaired mice because of their propensity to travel further than other mice, thus increasing their susceptibility to encountering both predators and parasites.

Line 395: "et al." need not be in italics here to remain consistent with the rest of the text

Line 398: comma after "study" and delete "on," or reword the sentence as I find it difficult to understand what you are trying to say.

Line 432: "developed"

Referee: 2

Comments to the Author(s)

Here, the authors use a powerful long-term dataset of wild *Peromyscus* to address a widespread convention that physical impairment impacts host fitness. They investigated limb, tail, and eye impairments on host survival, movement, weight, and ectoparasite infection and conclude that there is no support for impairments negatively impacting host movement & fitness save for

moderate effects of increased bot fly infection. I found this manuscript enjoyable to read and a robust & thoughtful study. I have some comments which I feel should be clarified/ addressed before publication. I detail these below but feel they are all addressable with minor revisions and that this study is well-suited to Proceedings B.

Minor comments

Lines 23-25 in the abstract highlight results which were not supported by the models, particularly the effects on ticks and persistence. I think the phrasing in the discussion which presents the results as similar persistence times between the groups and no effect on ticks is more transparent and that these lines as they are slightly misleading for the model results.

Line 99 – It would be helpful here just to clarify/confirm whether the recording of impairments was standard recording for notes at each capture. It seems so but initially I wondered if there could be any missing impairments if this was not an aim of data collection throughout all years trapped.

Lines 106-108 – Given the small sample size this is not much of an issue, but I wondered if authors could have three columns for tail, eye, or limb impairments as binary variables rather than having one variable for impairments and having to assign mice to one category?

Line 122-123 – Was the assumption of uniform capture probability between these two groups tested in any way? It seems that several mechanisms being tested in this study could also affect capture probability (eg movement/ fitness).

Lined 128-130 – The N of mice for which a status change of unimpaired to impaired over the course of trapping would be informative here.

Lines 138-140 – Does ‘selection criteria’ here refer only to age and week of first capture as mentioned further down in this section? If so suggest explicitly stating when first mentioning selection criteria.

Line 201 – Some justification for why the interaction of sex and season was not tested in ectoparasite models but was for other metrics tested would be useful.

Lines 201-203 – It is not clear to me from this description why negative binomial models were used for all (?) ectoparasites and then botflies were singled out for Poisson models for mice with at least one botfly.

Lines 222-223 – Clarify here what it indicates that the impaired population correlates with overall mouse population. In space? In time? Something else?

Lines 236-237 – As stated above this result comes as surprising based on the abstract. It is reasonable to present the raw data summaries throughout the results, in several instances it reads as a bit of an afterthought that there were no effects from the models accounting for other covariates, so would caution conservative phrasing of the raw data comparisons.

Lines 276-277 – This statement regarding selection bias is a bit lost on me in terms of conveying how this was determined, perhaps rephrase?

Line 316 – How was zero-inflation ruled out?

Line 330 – The use of ‘risk’ here seems in contrast following results describing effects on bot fly number as that term should pertain to probability of infection and not load, while there was no effect on odds of botfly infestation.

Lines 336-337 – Incidence rate seems inappropriate to describe data that does not include zeroes, as it suggests information about the risk/rate of cases, which requires there to be uninfected and infected individuals. Some clarification on the ‘at least one’ criteria may be needed if this does not refer to only non-zero data.

Line 405-406 – Unclear what ‘risk’ is referring to in this sentence given this is discussing body mass.

Statistical models – I had a few queries about the treatment of terms in statistical models.

1. Given the number of levels in the variable, it seems more suitable for year to be considered as a random effect.
2. It was unclear to me why grid was considered a random effect for some models but was a fixed effect for tick and movement models.
3. Was there a reason the intercept level for year changes between models for mass and movement?

Discussion – Just a general comment that I felt the authors handled the discussion very nicely and logically. I found the hypotheses regarding vertical movement and future directions very interesting and thought-provoking, particularly potentially implications of social interactions affected by impairments.

Author's Response to Decision Letter for (RSPB-2021-0886.R0)

See Appendix A.

RSPB-2021-1942.R0

Review form: Reviewer 1

Recommendation

Accept as is

Scientific importance: Is the manuscript an original and important contribution to its field?

Good

General interest: Is the paper of sufficient general interest?

Good

Quality of the paper: Is the overall quality of the paper suitable?

Excellent

Is the length of the paper justified?

Yes

Should the paper be seen by a specialist statistical reviewer?

No

Do you have any concerns about statistical analyses in this paper? If so, please specify them explicitly in your report.

No

It is a condition of publication that authors make their supporting data, code and materials available - either as supplementary material or hosted in an external repository. Please rate, if applicable, the supporting data on the following criteria.

Is it accessible?

Yes

Is it clear?

Yes

Is it adequate?

Yes

Do you have any ethical concerns with this paper?

No

Comments to the Author

You nailed it. Much much improved over the previous submission. This one was a pleasure to read. I have no further modifications.

Review form: Reviewer 2

Recommendation

Accept as is

Scientific importance: Is the manuscript an original and important contribution to its field?

Excellent

General interest: Is the paper of sufficient general interest?

Good

Quality of the paper: Is the overall quality of the paper suitable?

Excellent

Is the length of the paper justified?

Yes

Should the paper be seen by a specialist statistical reviewer?

No

Do you have any concerns about statistical analyses in this paper? If so, please specify them explicitly in your report.

No

It is a condition of publication that authors make their supporting data, code and materials available - either as supplementary material or hosted in an external repository. Please rate, if applicable, the supporting data on the following criteria.

Is it accessible?

Yes

Is it clear?

Yes

Is it adequate?

Yes

Do you have any ethical concerns with this paper?

No

Comments to the Author

I would like to commend the authors for their thorough response to reviews. They have carefully responded to all comments raised in my previous review, and I particularly appreciate the revision of the statistical models to address any concerns as I know this is time-consuming. I have no remaining issues and think this will make a very interesting contribution to the literature.

Decision letter (RSPB-2021-1942.R0)

04-Oct-2021

Dear Ms Rubino

I am pleased to inform you that your manuscript RSPB-2021-1942 entitled "Effects of Physical Impairments on Fitness Correlates of the White-footed Mouse, *Peromyscus leucopus*" has been accepted for publication in Proceedings B.

The referees and the Associate Editor have recommended publication, but the Associate Editor suggests some minor revisions to your manuscript. Therefore, I invite you to respond to the comments and revise your manuscript. Because the schedule for publication is very tight, it is a condition of publication that you submit the revised version of your manuscript within 7 days. If you do not think you will be able to meet this date please let us know.

- 1) A text file of the manuscript (doc, txt, rtf or tex), including the references, tables (including captions) and figure captions. Please remove any tracked changes from the text before submission. PDF files are not an accepted format for the "Main Document".
- 2) A separate electronic file of each figure (tiff, EPS or print-quality PDF preferred). The format should be produced directly from original creation package, or original software format. PowerPoint files are not accepted.

3) Electronic supplementary material: this should be contained in a separate file and where possible, all ESM should be combined into a single file. All supplementary materials accompanying an accepted article will be treated as in their final form. They will be published alongside the paper on the journal website and posted on the online figshare repository. Files on figshare will be made available approximately one week before the accompanying article so that the supplementary material can be attributed a unique DOI.

Sincerely,

Professor Hans Heesterbeek

Associate Editor

Board Member

Comments to Author:

The authors have performed a thorough revision of their manuscript in response to two expert reviews. As a result the manuscript is much improved and the authors deserve credit for their careful consideration of all concerns raised previously by the reviewers. A few small issues remain. 1) It appears that the authors may have accidentally deleted the abstract and keywords from the submitted version of the revised manuscript? At least these are missing from the version available through the submission portal. 2) In their revisions to the abstract in response to reviewer 2, if the new statement "Mice with impairments were roughly 5% heavier and had comparable persistence times as well as burdens of black-legged ticks (*Ixodes scapularis*) and bot fly larvae (*Cuterebra*) compared to mice without impairments" refers to mass and not weight, please see the comment from reviewer 1 to use the terms "greater mass" and "less mass" instead of "lighter" and "heavier" throughout the manuscript.

Reviewer(s)' Comments to Author:

Referee: 1

Comments to the Author(s).

You nailed it. Much much improved over the previous submission. This one was a pleasure to read. I have no further modifications.

Referee: 2

Comments to the Author(s).

I would like to commend the authors for their thorough response to reviews. They have carefully responded to all comments raised in my previous review, and I particularly appreciate the revision of the statistical models to address any concerns as I know this is time-consuming. I have no remaining issues and think this will make a very interesting contribution to the literature.

Author's Response to Decision Letter for (RSPB-2021-1942.R0)

See Appendix B.

Decision letter (RSPB-2021-1942.R1)

11-Oct-2021

Dear Ms Rubino

I am pleased to inform you that your manuscript entitled "Effects of Physical Impairments on Fitness Correlates of the White-footed Mouse, *Peromyscus leucopus*" has been accepted for publication in Proceedings B.

Your article has been estimated as being 9 pages long. Our Production Office will be able to confirm the exact length at proof stage.

Data Accessibility section

Open Access

Paper charges

Sincerely,

Appendix A

Response to Referees for Manuscript: RSPB-2021-0886

Associate Editor

Comments to Author:

Following two expert reviews there is considerable enthusiasm that the manuscript submitted by Rubino et al will make a significant contribution to understanding how physical impairment affects fitness in wildlife. The reviewers raise a few issues that should be addressed in any revision. One reviewer raises the point that the comparison to the human eugenics movement, while an interesting hook for the article, is a bit of a stretch and therefore the manuscript should focus on physical impairment in wildlife as the motivating framework for the research. Additionally, an interesting point is raised concerning cause and effect: do individuals with impairments have larger home ranges because of impairment, or would unimpaired individuals be more likely to become impaired due to having larger home ranges? Minor concerns also were raised about the heavy reliance on supplementary tables, which most readers are unlikely to read, and the need for more detailed statistical reporting in the main text. The authors should address all of the points raised by the reviewers in preparing a revision of their manuscript.

Response: We thank the Associate Editor for these comments. As we detail below in responses to specific reviewer comments and suggestions, we have carefully revised the manuscript to address these two major comments, regarding the comparison to human eugenics theory and the potential for reverse causality in the case of impairments and home range size. In addition, we have changed the balance of supplementary tables and the main text as repositories of the results, and addressed all other concerns raised in the reviews.

Reviewer(s)' Comments to Author:

Referee: 1

Comments to the Author(s)

This paper is well written, and is generally a contribution to the literature, but needs some work.

Response: We thank the reviewer for this assessment and have undertaken considerable work to improve the manuscript.

Line 24: need to include Latin binomial or at least genus for “tick and bot fly”

Response: We have added the Latin binomial for the tick and the genus for the bot fly [line 27]. In the case of the latter, a species identification could not be confirmed on the basis on morphological characteristics.

Line 36: “Natural Selection” need not be capitalized

Response: We have removed this sentence.

Line 41: comma after “thus”

Response: We have removed this sentence.

Lines 36-46: The title and abstract mention and specifically speak about mice. I think when talking about “individuals” in these opening lines, you need to make clear you are talking about humans. Lines 41 and 44, change “individuals” to “humans.”

Response: We have modified this section to make use of the word “individuals” clearer. As noted below, we have strongly de-emphasized the focus on humans, resulting in a rewrite of these lines [lines 38-42].

Lines 47-48: Do you really think, and expect the readership to believe, that the role impairments play in humans can be better understood by looking at those in a single mouse population found on Cary Institute property? I would say this is a bit of stretch. And you are assuming that mice are born with these impairments and not sustaining them from predator encounters or from human sampling events to be able to compare to human birth defects? This opening paragraph is impossibly broad, comparing human eugenics to a mouse who may have lost its tail which saved its life in an attack from a barred owl. Your Abstract is spot on, using these data to attribute survival of the fittest within species is an addition to the literature base; pulling from and comparing to human eugenics is an enormous over-reach.

Response: We appreciate the perspective provided by the reviewer and agree that greater care is required in applying these results from six mouse populations to

humans. We have revised the paragraph to narrow the focus and removed discussion of the theory of natural selection and its influence on the devaluation of individuals to better focus on the specific implications of impairment in wild mice, which is the focus of the paper. We retained only one clause in the second sentence that mentions eugenic movements, providing citations. The context here is the broad impact arising from perceptions of the consequences of impairments in animals. We hope this revision will be considered more suitable.

Line 92: oak and maple should at least have the genus identified

Response: We have added the scientific names here [line 89].

Lines 94-95: I know this is a brief description and you reference more specific methods, but your description here seems to indicate you trapped mice every 3-4 weeks since 1991. I am guessing you were not trapping in January, so might want to include that you were trapping every 3-4 weeks from May – October, or whatever the range of months that might be.

Response: We appreciate this suggestion and have changed sentence [lines 91-92] to: “In brief, trapping was conducted every 3-4 weeks over 2-3 consecutive days between May and November using Sherman live traps baited with oats.”

Line 159: Likely not many mice under 11 grams existed in the data set because mice below that mass were unlikely to trigger Sherman traps.

Response: In fact, there were 2,387 entries within our dataset recording mice that were below 11 grams. But these were clearly juvenile animals and we excluded them from analyses focusing on adults [line 159].

Lines 197-198, 202, 213, 318, 320, 321, 325, 327, 328, 334, 336, 340, 350, 371, 375, 378, 382: need to specify they were parasitized by bot fly larvae, not bot flies

Response: The language “bot flies” has been replaced throughout with “bot fly larvae” [lines 27, 201, 213, 314, 319, 322, 335, 372, 377].

Line 202: why is “botfly” one word here and two words everywhere else?

Response: The word has been corrected to “bot fly.”

Line 238: “3” should be “3.0”

Response: The next now states 3.0.

Line 243: “3096” should be “3,096” to remain consistent with previous usage

Response: The comma has been added [line 232].

Line 247: comma after “average”

Response: A comma has been added [line 242].

Line 247: “lighter” is a reference to weight. Here you are measuring mass. “On average, adult mice without impairments had less mass . . .”

Response: The term “lighter” has been removed so that we now refer to “less mass” [line 242].

Line 248: delete the comma and insert “mass” after “observed”

Response: The grammar has been corrected [line 243].

Lines 249-250: don't you need to swap the order of limb impaired mice and eye impaired mice if you are reporting those cohorts in descending order of average mass?

Response: The order has been corrected [lines 243-245].

Lines 251-261: You keep saying/implying that various impairments caused an increase in mass, but that is really not the case. Mice with these impairments had more mass, generally. Semantics, I know. And this is not surprising as elder mice (with more mass) will have had more intraspecific and interspecific conflicts than younger mice merely as a function of age.

Response: We appreciate these comments. In these descriptive sections of the results, we were not attempting to infer causality, but simply trying to state the nature of the association between impairment and mass (as well as the various other response variables). To clarify this, we have changed the language [lines: 246-256] from <x impairment was associated with y increase> to having <x impairment was associated with having on average y grams more mass>. For example: “tail impairments were significantly associated with having on average 1.43g more mass”.

Lines 262-263: You had multiple leucopus with 30-50 grams of mass in your study? In the thousands we have trapped over 15 years, we seldom have had one over 30 grams, and usually they were gravid females. Based on personal experience, I would have to agree with this site on the physical description of leucopus, including their mass of between 15-25 grams (https://animaldiversity.org/accounts/Peromyscus_leucopus/). I would double check these figures before publishing. They seem like they could be data errors in the “Absent, Missing Tail, and Partial Tail” box and whisker plots. Those are BIG leucopus that I have never seen reported in the literature previously. People in the know are going to question the veracity of your data if these data are published as is and are not addressed.

Response: We appreciate the reviewer noticing the details of mouse body mass. In the entire data set, there were 27 mice recorded as above 40 grams. All these records have now been excluded from the analysis based on the possibility that these records represent misreads of the Pesola scale (line 159 has been added to address this). This exclusion did not change our results in any meaningful way. We typically trap mice in the 30-40 grams range during late-stage pregnancy on Cary Institute grounds. Geographic variation in mass has been documented for this species (Nolfo-Clements et al.) and may be why we have different ranges in mass than in other regions where *P. leucopus* are studied. There are reports of mice in this same range of mass (<https://www.dept.psu.edu/nkbiology/naturetrail/speciespages/whitefootedmouse.htm>).

Lines 307-308: if you are going to include the mean ectoparasite burden for mice without impairments, you need to include it for impaired mice as well instead of just referring to the supplemental table. Do you have any statistical proof that mean burdens on impaired mice were significantly higher than without impairments? If so, here is where you need to include that in text. Otherwise, this is meaningless.

Response: We thank the reviewer for this suggestion. Descriptive statistics on ectoparasite burden for impaired mice have now been added to the main text; however, all descriptive statistics in this section were changed from means to medians to best describe the non-Gaussian nature of the data. Additionally, the lack of a statistical association is now clearly stated. Language has been changed in lines 302-307 to: “The median larval tick burden among mice was 2 for those without visible physical impairments (IQR:0-7) and those with either tail (IQR:0-9) or limb impairments (IQR:0-9) and 4 for those with eye impairments (IQR:0-10). After controlling for season, sex their interaction, and age, no significant associations were detected between impairments and the odds of tick infestation (Tail: OR=0.99, 95% CI: 0.84-1.16, p=0.87; Limb: OR=0.91, 95% CI: 0.58-1.42, p=0.68; Eye: OR=0.87, 95% CI: 0.51-1.50, 0.62). However, higher tick loads were associated with tail impairments after controlling for the same variables (Tail: IRR=1.11, 95% CI: 1.01-1.22, p=0.04). No association between tick load and eye or limb impairments overall were detected (Limb: OR=1.14, 95% CI: 0.89-1.47, p=0.30; Eye: OR=1.19, 95% CI: 0.88-1.61, 0.26). A slight association between a higher tick load and missing a limb was observed in the model analysing specific impairments (IRR=1.51, 95% CI: 0.99-2.29, p=0.05).”

Line 310: comma after “average”

Response: We have added the comma.

Lines 312-314 and 314-316: even if there was no significance, you need to include statistical outputs here.

Response: Statistical outputs have been added to lines 304-307 and 309-311. As stated above:

“After controlling for season, sex their interaction, and age, no significant associations were detected between impairments and the odds of tick infestation (Tail: OR=0.99, 95% CI: 0.84-1.16, p=0.87; Limb: OR=0.91, 95% CI: 0.58-1.42, p=0.68; Eye: OR=0.87, 95% CI: 0.51-1.50, 0.62). However, higher tick loads were associated with tail impairments after controlling for the same variables (Tail: IRR=1.11, 95% CI: 1.01-1.22, p=0.04). No association between tick load and eye or limb impairments overall were detected (Limb: OR=1.14, 95% CI: 0.89-1.47, p=0.30; Eye: OR=1.19, 95% CI: 0.88-1.61, 0.26).”

Lines 318-319: need to include mean burdens in text here as well instead of referencing a supplemental table

Response: The following language on percent infested has been added [lines 315-319] (the data are highly skewed and thus means are not reported in the text, nor medians as the median across all groups was zero): “During the bot fly season the number of bot fly larvae per mouse ranged from 0-9 over the study period. The percent of captures infested with at least one bot fly ranged from 11.11% of mice with limb impairments, 12.44% of mice who became impaired at a later date, 13.05% of mice without any visible physical impairments recorded, 17.11% of mice with tail impairments, and 18.63% of mice with eye impairments (Table S14).

Line 319-320, 321-323: need to include statistical output showing lack of significant differences

Response: Statistical output has now been added [lines 323-326]: “After controlling for season, sex, and age in the logistic mixed-effects models, a significant association was observed between the odds of having at least one bot fly larvae and having an eye impairment (OR=1.81; 95% CI: 1.01-3.25; p=0.05). More specifically, mice with cataracts had 2.19 times the odds (95% CI: 1.04-4.62) as mice without visible physical impairments of being infested with bot fly larvae (p=0.04). No significant association was detected for a tail or limb impairment, nor among mice who became impaired at a later date, summarized in Tables S15-16 (Tail: OR=1.08, 95% CI: 0.88-1.31, p=0.46; Limb: OR=0.89, 95% CI: 0.48-1.63, p=0.70; Future Impaired: OR=0.86, 95% CI: 0.65-1.12, 0.26).”

Lines 356-365: You have this exactly backwards. The mice do not have larger home ranges because they had a tail impairment, they have tail impairments because they

had larger home ranges. These mice prior to becoming impaired, were pre-programmed to wander and have larger home ranges, placing them further away from the safety of their dens than their counterparts. As a result, these mice were more exposed and more likely to encounter a predator attack, thus causing said tail impairment. Most mice with larger home ranges without tail impairments were dead as the result of predation and were not available to be trapped. This entire paragraph needs to be omitted and re-written because it is wrong and will fall completely flat if published as is.

Response: We agree that our data comparing MSD of impaired mice (pre-impairment and overall) with that of unimpaired mice are consistent with the mechanism proposed by the reviewer. Consequently, this section has been omitted.

Line 367-369: Yes!!! Omit the vertical habitat usage altogether and focus on this. Mice with larger home ranges are more exposed and thus, more at risk. This is the focus of this paragraph (not a closing example) and not that they cannot climb because of a damaged tail.

Response: Our new paragraph [354-360] omits the earlier section and now reads: "The larger home ranges of mice with impairments compared to mice without impairments suggests that their horizontal movement was not impeded by their impairments. The larger home ranges may be linked to behavioural differences in the impaired versus non-impaired mice. For instance, if impaired mice are less risk averse, they may use more space in daily activities; this behaviour may have also increased their risk of becoming injured from predators or agonistic intraspecific interactions which led to their impairment."

Line 372: period after "al"

Response: A period has been added.

Line 377: "...may come into more frequent contact with bot fly eggs."

Response: The grammar in this sentence has been corrected; it now reads: "Mice with eye impairments may spend more time on the ground where they may come into more frequent contact with bot fly eggs."

Line 381: delete second "in"

Response: The word "in" has been deleted; the sentence now reads: "Stressful stimuli are associated with an increase in parasite burden in many species [48–53] including *Peromyscus leucopus* [54]."

Line 383: In lines 20 and 89, you mention you monitored a single wild population of white-footed mice. Why here then are you suggesting you monitored multiple populations?

Response: We monitored mouse populations on six different trapping grids, with virtually no mice being caught in more than one of these grids. Hence, we consider them as multiple populations, for purposes of describing our sampling regimen. We now state [lines 20 and 388] that we are referring to field populations, plural, to reflect this situation.

Line 391: It is likely you saw higher parasitism on impaired mice because of their propensity to travel further than other mice, thus increasing their susceptibility to encountering both predators and parasites.

Response: The second sentence in this section has been changed to: “The larger home ranges of mice with impairments compared to those without impairments may have increased the probability of encountering host-seeking ticks (which are sedentary and rely on hosts to approach them).”

Line 395: “et al.” need not be in italics here to remain consistent with the rest of the text

Response: Italics have been removed.

Line 398: comma after “study” and delete “on,” or reword the sentence as I find it difficult to understand what you are trying to say.

Response: A comma was added, and “on” was removed; the sentence now reads: “The role intra-specific social interactions play in the success of impaired individuals was beyond the scope of this study, but likely influences the implications of these findings to other organisms.”

Line 432: “developed”

Response: This correction has been made; “develops” has been changed to “developed”

Referee: 2

Comments to the Author(s)

Here, the authors use a powerful long-term dataset of wild *Peromyscus* to address a widespread convention that physical impairment impacts host fitness. They investigated limb, tail, and eye impairments on host survival, movement, weight, and ectoparasite infection and conclude that there is no support for impairments negatively impacting host movement & fitness save for moderate effects of increased bot fly infection. I found this manuscript enjoyable to read and a robust & thoughtful study. I have some comments which I feel should be clarified/ addressed before publication. I detail these below but feel they are all addressable with minor revisions and that this study is well-suited to Proceedings B.

Response: We very much appreciate the positive comments by the reviewer and have made every effort to address the comments, as described below.

Minor comments

Lines 23-25 in the abstract highlight results which were not supported by the models, particularly the effects on ticks and persistence. I think the phrasing in the discussion which presents the results as similar persistence times between the groups and no effect on ticks is more transparent and that these lines as they are slightly misleading for the model results.

Response: The descriptive results have been removed, and the sentence now reads: “Mice with impairments were roughly 5% heavier and had comparable persistence times as well as burdens of black-legged ticks (*Ixodes scapularis*) and bot fly larvae (*Cuterebra*) compared to mice without impairments”

Line 99 – It would be helpful here just to clarify/confirm whether the recording of impairments was standard recording for notes at each capture. It seems so but initially I wondered if there could be any missing impairments if this was not an aim of data collection throughout all years trapped.

Response: We addressed this concern by adding the following statement [lines 97-98]: “Although this was not the main aim of the data collection, established protocols instructed all trappers to record detailed notes about physical features of each mouse in the notes, including impairments.”

Lines 106-108 – Given the small sample size this is not much of an issue, but I wondered if authors could have three columns for tail, eye, or limb impairments as binary variables rather than having one variable for impairments and having to assign mice to one category?

Response: We appreciate this concern. We did not wish to compare mice with one type of impairment to all other mice (thus including those with the other two types of impairments) as this could reduce observed differences between mice with and without impairments.

Line 122-123 – Was the assumption of uniform capture probability between these two groups tested in any way? It seems that several mechanisms being tested in this study could also affect capture probability (eg movement/ fitness).

Response: We apologize for imprecise language here. We have changed the language to better reflect our original intention which was that capture probability was assumed to be similar, but we are not making assumptions about individual heterogeneities beyond that. Uniform has been changed to similar.

Lined 128-130 – The N of mice for which a status change of unimpaired to impaired over the course of trapping would be informative here.

Response: N=301 has been added to the end of the sentence.

Lines 138-140 – Does ‘selection criteria’ here refer only to age and week of first capture as mentioned further down in this section? If so suggest explicitly stating when first mentioning selection criteria.

Response: The following has been added to this sentence “(age, week of first capture, and minimum persistence as described below)”.

Line 201 – Some justification for why the interaction of sex and season was not tested in ectoparasite models but was for other metrics tested would be useful.

Response: Thank you for the suggestion. We have tested the interaction between sex and season for the ectoparasite models and added this term to both the sentence in the methods and model results where needed.

Lines 201-203 – It is not clear to me from this description why negative binomial models were used for all (?) ectoparasites and then botflies were singled out for Poisson models for mice with at least one botfly.

Response: We thank the reviewer for identifying this issue. Bot flies are much rarer than the other ectoparasites (i.e. ticks) we analyzed, with infestation intensities of zero through 3-4 larvae per mouse, so we originally decided a priori that Poisson models might be preferable. After reanalyzing the data, we have concluded that a negative binomial, zero-truncated model is more appropriate (and a better fit) for all the ectoparasite data. Now all of the models for ectoparasite load are negative binomials.

Lines 222-223 – Clarify here what it indicates that the impaired population correlates with overall mouse population. In space? In time? Something else?

Response: We have now clarified that we mean over time. The sentence now reads: “The impaired mice population correlated significantly with the overall mouse population over time ($r(24)= 0.86, P<0.001$) (Figure S1).”

Lines 236-237 – As stated above this result comes as surprising based on the abstract. It is reasonable to present the raw data summaries throughout the results, in several instances it reads as a bit of an afterthought that there were no effects from the models accounting for other covariates, so would caution conservative phrasing of the raw data comparisons.

Response: The sentence on median survival times has been removed to enable the focus to remain on the model results. The reference to the table and figure containing raw data summaries is now stated in the newly added sentence: “No statistically significant differences in median survival times (Table S3 and Figure 1) were detected between impaired and non-impaired mice ($X^2=2.3, P= 0.1$) nor across impairment types and the non-impaired mice ($X^2=3.0, P= 0.4$).”

Lines 276-277 – This statement regarding selection bias is a bit lost on me in terms of conveying how this was determined, perhaps rephrase?

Response: We thank the reviewer for pointing out this ambiguity. The word “selection” was removed.

Line 316 – How was zero-inflation ruled out?

Response: The following sentence has been added to the methods, lines 203-204: “Zero-inflation was ruled out a priori and confirmed as appropriate post-hoc using the Vuong test.”

Line 330 – The use of ‘risk’ here seems in contrast following results describing effects on bot fly number as that term should pertain to probability of infection and not load, while there was no effect on odds of botfly infestation.

Response: Thank you for the correction; “increased risk” has been replaced with “higher load.”

Lines 336-337 – Incidence rate seems inappropriate to describe data that does not include zeroes, as it suggests information about the risk/rate of cases, which requires there to be uninfected and infected individuals. Some clarification on the ‘at least one’ criteria may be needed if this does not refer to only non-zero data.

Response: Incidence rate has been replaced by “load.” This sentence has been removed from the results, as all three negative binomial models now produce

quite similar results. We have reduced the prior sentences on these models to: “The negative binomial models showed similar results for bot fly load after controlling for these same factors (Tables 3 and S16-17).”

Line 405-406 – Unclear what ‘risk’ is referring to in this sentence given this is discussing body mass.

Response: The sentence has been corrected to read: “As no statistically significant change in mass was detected for any of the impairments ...”

Statistical models – I had a few queries about the treatment of terms in statistical models.

1. Given the number of levels in the variable, it seems more suitable for year to be considered as a random effect.

Response: We thank the reviewer for the suggestion. Year is now treated as a random effect in each model.

2. It was unclear to me why grid was considered a random effect for some models but was a fixed effect for tick and movement models.

Response: Grid is now treated as a random effect in both the tick and movement models. Although there are valid reasons why grid could be treated either as a fixed or random effect in these models, the treatment of the variable does not influence the results concerning the associations of interest (i.e., the relationship between impairment and ticks/movement). We therefore decided to treat grid as a random effect throughout the paper, as suggested by the reviewer. “Grid” is now referred to as “Plot” to distinguish the trapping locations from the use of the word “grid” in the methodology portion of the movement analysis.

3. Was there a reason the intercept level for year changes between models for mass and movement?

Response: The reference year was selected based upon which year had the greatest sample size for the particular outcome of interest, which was slightly greater in 2007 for MSD and in the 2016 for mass (due to subsetting for animals with sufficient data for each analysis). This is no longer an issue as year is now treated as a random variable.

Discussion – Just a general comment that I felt the authors handled the discussion very nicely and logically. I found the hypotheses regarding vertical movement and future directions very interesting and thought-provoking, particularly potentially implications of social interactions affected by impairments.

Response: We sincerely appreciate the suggestions, corrections, and thoughtful critique.

Appendix B

Response to Referees for Manuscript: RSPB-2021-0886

Associate Editor

Board Member

Comments to Author:

The authors have performed a thorough revision of their manuscript in response to two expert reviews. As a result, the manuscript is much improved, and the authors deserve credit for their careful consideration of all concerns raised previously by the reviewers. A few small issues remain. 1) It appears that the authors may have accidentally deleted the abstract and keywords from the submitted version of the revised manuscript? At least these are missing from the version available through the submission portal. 2) In their revisions to the abstract in response to reviewer 2, if the new statement “Mice with impairments were roughly 5% heavier and had comparable persistence times as well as burdens of black-legged ticks (*Ixodes scapularis*) and bot fly larvae (*Cuterebra*) compared to mice without impairments” refers to mass and not weight, please see the comment from reviewer 1 to use the terms “greater mass” and “less mass” instead of “lighter” and “heavier” throughout the manuscript.

Response: We thank the Associate Editor for these comments. We have addressed both issues regarding the abstract and keywords. The abstract and keywords have been added back into the revised manuscript (1). We have revised the line (24) in the manuscript corresponding to the statement about mass to read: “Mice with eye and tail impairments had 5% and 6% greater mass, respectively, than unimpaired mice.”

Reviewer(s)' Comments to Author:

Referee: 1

Comments to the Author(s).

You nailed it. Much, much improved over the previous submission. This one was a pleasure to read. I have no further modifications.

Response: We thank the reviewer for their assessment, thoughtful critique, and the time and effort they contributed to improving the manuscript.

Referee: 2

Comments to the Author(s).

I would like to commend the authors for their thorough response to reviews. They have

carefully responded to all comments raised in my previous review, and I particularly appreciate the revision of the statistical models to address any concerns as I know this is time-consuming. I have no remaining issues and think this will make a very interesting contribution to the literature.

Response: We thank the reviewer for their advice on improving the statistical models and their commendation; we appreciate the time and effort they contributed to improving the manuscript.